**Data Availability Statement:** Data is available on Kaggle. The corresponding DOI is: 10.34740/kaggle/dsv/1396507 (https://doi.org/10.34740/kaggle/dsv/1396507). Alternative URL: https://

# Subjective ratings of emotive stimuli predict the impact of the COVID-19 quarantine on affective states

**Héctor López-Carral**[1,2☯], **Klaudia Grechuta**[1☯], **Paul F. M. J. Verschure**[1,3]*

**1** Laboratory of Synthetic, Perceptive, Emotive and Cognitive Systems (SPECS), Institute for Bioengineering of Catalonia (IBEC), The Barcelona Institute of Science and Technology (BIST), Barcelona, Spain, **2** Universitat Pompeu Fabra (UPF), Barcelona, Spain, **3** Institució Catalana de Recerca i Estudis Avançats (ICREA), Barcelona, Spain

☯ These authors contributed equally to this work.
* pverschure@ibecbarcelona.eu

## Abstract

The COVID-19 crisis resulted in a large proportion of the world's population having to employ social distancing measures and self-quarantine. Given that limiting social interaction impacts mental health, we assessed the effects of quarantine on emotive perception as a proxy of affective states. To this end, we conducted an online experiment whereby 112 participants provided affective ratings for a set of normative images and reported on their well-being during COVID-19 self-isolation. We found that current valence ratings were significantly lower than the original ones from 2015. This negative shift correlated with key aspects of the personal situation during the confinement, including working and living status, and subjective well-being. These findings indicate that quarantine impacts mood negatively, resulting in a negatively biased perception of emotive stimuli. Moreover, our online assessment method shows its validity for large-scale population studies on the impact of COVID-19 related mitigation methods and well-being.

## Introduction

In December 2019, Chinese health authorities reported a cluster of pneumonia cases in the city of Wuhan, in the Hubei province, caused by the novel coronavirus SARS-CoV-2 (COVID-19) [1]. By mid-March 2020, a total of 200,000 confirmed cases [2] had been reported worldwide, showing an exponential increase with the current number of identified cases exceeding 14 million, whereby Spain, Italy, and the United Kingdom are the most-affected European nations.

To prevent the spread of COVID-19, public health authorities have employed mitigation strategies and, in particular, quarantine [3] and isolation, which are currently practiced across the globe. Mandatory mass quarantine restrictions, which include social distancing, stay-at-

www.kaggle.com/hectorlopezcarral/covid19-affective-ratings.

**Funding:** This research has been supported by the European Commission under contract H2020-787061 (ANITA) and H2020-840052 (cRGS), and by EIT Health under grant ID 19277 (RGS@home) to PFMJV.

**Competing interests:** The authors of this manuscript have the following competing interests: PFMJV is the founder and interim CEO of Eodyne S.L., which aims at bringing scientifically validated neurorehabilitation technology to society. This does not alter our adherence to PLOS ONE's policies on sharing data and materials. The rest of the authors have nothing to disclose.

home rules, and limiting work-related travel outside the home [4] might impact both physical and mental health of the affected individuals [5]. Indeed, prolonged widespread lock-down and limiting social contact has resulted in post-traumatic stress disorder, depression, anxiety, mood dysregulations, and anxiety-induced insomnia during previous periods of quarantine [6–8]. These, in turn, led to cognitive distortions and maladaptive behaviors, including suicide [9, 10]. A growing body of evidence from COVID-19 demonstrates that the current mass quarantine has been producing similar adverse psychological effects, which might have long-lasting consequences on both individual subjects and society [5, 11–13]. Moreover, it is unclear for how long and how frequent confinement measures will be put in place in the medium and long-term. Hence, understanding the specific impact of COVID-19 on mental health and the development of monitoring and diagnostic tools to identify individuals at risk are of critical importance.

Disturbances in mental health, including disorders of mood, are commonly assessed using explicit questionnaires and interview measures [14]. Both clinician-rated and self-reported instruments have been used for decades [15]. Some studies, however, have outlined noteworthy limitations of standard assessments of depression, such as conceptual and psychometric flaws [16–20]. For instance, the Hamilton Depression Rating Scale (HDRS, [21]), which has been considered a gold standard in clinical practice as well as clinical trials, was widely criticized for its subjectivity as well as the multidimensional structure, which varies across studies hence preventing replication across samples as well as poor factorial and content validity [16–20, 22]. Moreover, it is well-established that self-reports in psychological research can suffer from response bias such as socially desirable responding or a tendency to provide positive self-descriptions [23–25]. To counteract possible response bias and suggestion effects, in the current study, we employed affective ratings of calibrated emotional stimuli as an implicit measure of mental state building on earlier validation studies of online emotional rating methods of calibrated emotional stimuli [26].

Mood-state-dependent changes in emotional reactivity are reflected in emotion experience evaluations [27]. Indeed, there is converging evidence that ratings of affective stimuli might serve as a robust, indirect measure of mood. For example, empirical studies show reduced subjective and expressive emotional responses to neutral and positive stimuli in depression, including in major depressive disorder (MDD) [28–31]. Specifically, the results show significant negative shifts in emotional ratings of valence compared to the healthy controls such that patients judge the stimuli as substantially less pleasant. Alternatively, Borderline Personality Disorder (BPD) patients show hypersensitivity to emotional stimuli as compared to healthy controls [32]. These findings support the notion that response to emotive stimuli is be altered in disorders of mood.

Given the mental health risk of medium to long-term isolation [7, 8, 33], it is relevant to develop methods that can effectively and unobtrusively assess and monitor the impact of the restriction of movement and social distancing on well-being and mental health. Hence, the goal of this study is to evaluate the effects of quarantine-induced changes in mood, as measured implicitly through the subjective ratings of emotional stimuli. We predicted that individuals in quarantine due to COVID-19 might present changes in their affective ratings that reflect their subjective experience of isolation. To test this hypothesis, we conducted an online experiment in which volunteers were asked to rate the affective content of a subset of standardized visual stimuli and report their current personal situation and experience related to the pandemic. We compared the affective ratings of valence (i.e., indicative of disturbances in mood) between groups of subjects in the pre-quarantine "normal" condition and under quarantine.

## Materials and methods

### Participants

After providing their consent, one hundred twelve subjects participated in the study (64.29% females) with a mean age of 32.38 ($SD$ = 9.04). The sample size of $N$ = 110 was determined a priori using G*Power software version 3.1 (Kiel, Germany) based on $\alpha$ = 0.05, power of 80% and medium effect size (0.5). Volunteers accessed the online experiment using a URL (uniform resource locator) that was shared through social media and instant messaging platforms by the experimenters. 51.79% of the subjects held postgraduate degrees or higher. Subjects originated from 19 different countries (30.36% Spanish and 21.43% Italian), and they lived in 17 countries (53.57% in Spain and 16.07% in Italy). This sampling approach was chosen to cover a range of countries that were similarly impacted by self-isolation measures. In particular, for the analyses, we included only those participants who were actively undergoing quarantine. Thus, all participants were uniform in their cultural traits [34] and quarantine measures, including social isolation and distancing, the banning of social events and gatherings, the closure of schools, offices, and other facilities, and travel restrictions [35, 36].

The reported data were collected between the 9th and the 20th of April 2020. The personal data of the subjects were anonymized and kept confidential. All participants were blind to the purpose of the study. Specifically, until the end of the session, subjects did not know the study's objective, which could bias their responses. However, they were informed about it at the end of the trial.

### Materials

**Affective Slider.**   We employed the Affective Slider tool [26] for digital assessments of the arousal and pleasure dimensions of the emotive stimuli. Its design principles follow the circumplex model of emotion proposed by James Russell [37, 38]. In this bipolar model, arousal corresponds to the intensity of an affective response (i.e., evoked level of excitement), while valence represents the positivity or negativity of the response (i.e., happiness). Consequently, the Affective Slider consists of a pair of slider controls flanked by emoticons that correspond to the ratings of arousal and valence, respectively. Both sliders are oriented horizontally and located above each other (Fig 1). In this study, Affective Slider served to allow for continuous subjective assessment of the presented images, thus counteracting methodological limitations of classical scales such as the Self-Assessment Manikin (SAM) [39] especially when applied in online assessments [26]. During the experiment, the position of the two sliders on the screen

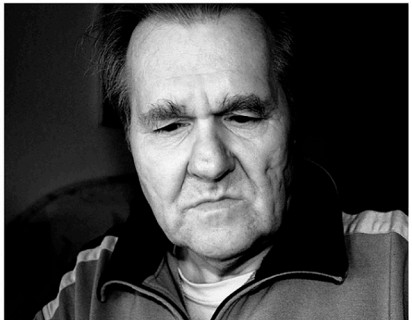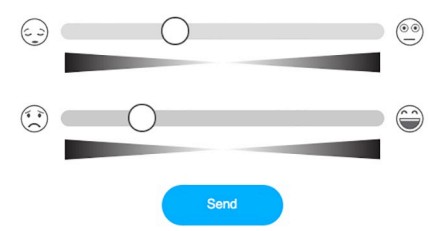

**Fig 1. Example of digital assessments of the arousal and pleasure using the Affective Slider [26].** On the left, there is an example image from the OASIS data set [40]. On the right, there are the ratings. The top slider corresponds to arousal and the bottom one to valence. This visual order was randomized over trials.

(e.g., arousal on top of valence or vice versa) randomly changed at every trial to prevent the order-effects and automaticity in the responses.

**Experimental stimuli: Open Affective Standardized Image Set (OASIS).** OASIS is a validated open-access data set, which consists of nine hundred images acquired online [40]. Each stimulus includes normative ratings of both arousal and valence reported on a scale between 1 and 7 by 822 participants. The stimuli depict a variety of themes within four categories that include people, animals, scenes, and objects. In contrast with the well-known International Affective Picture Set (IAPS) [41], OASIS allows for online use of the data set and provides more recent ratings. For the purpose of this study, we chose a subset of 30 images from the categories *people* and *scenes*, corresponding to 61.78% of the entire set. The choice was determined by the content of the stimuli, which was related to social and outdoor activities. The subset was selected randomly from the whole set of images to achieve a representative sample (see Fig 2). The same set of 30 images was presented to all participants in a randomized order (S1 File Image Selection).

**COVID-19 questionnaire.** To evaluate the current personal and social situation of each participant and their subjective experience during the COVID-19 global health crisis, we created a custom questionnaire. The scale was composed of 14 items, including an optional field

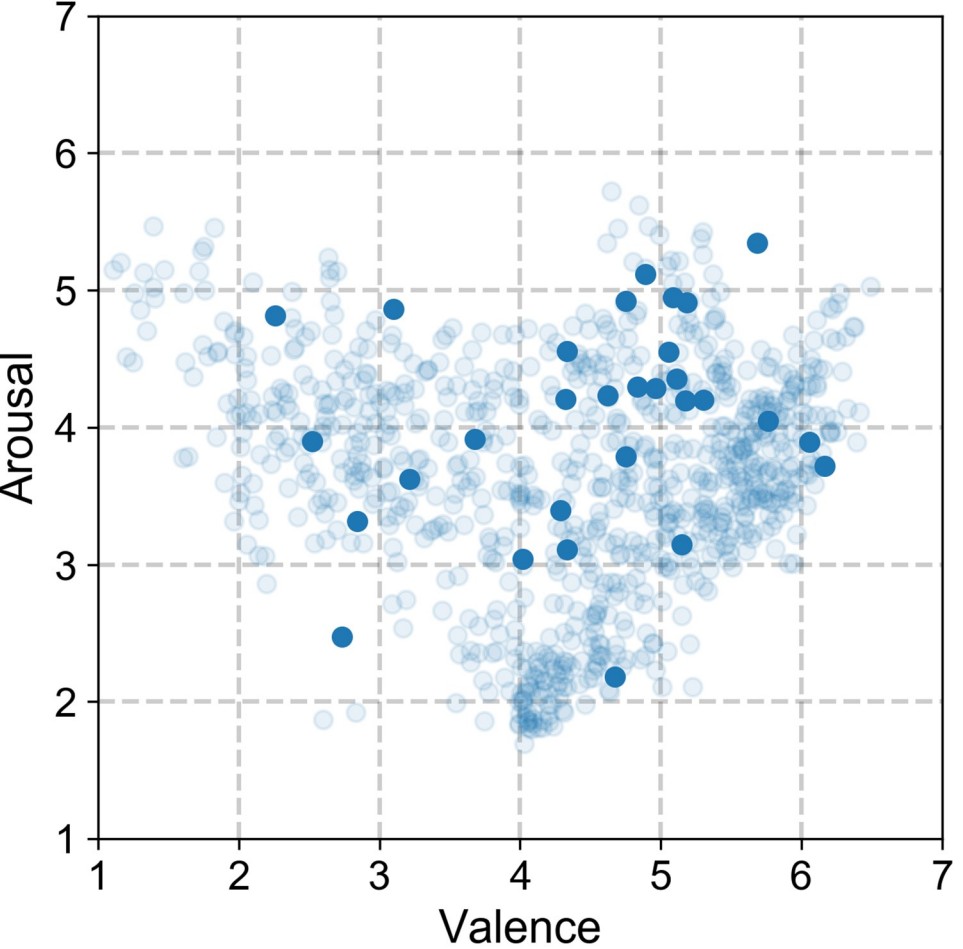

**Fig 2. Distribution of the valence and arousal rating for the 30 images selected for this study (solid circles) and the OASIS data set of 900 images (semitransparent circles).**

to provide personal comments related to the quarantine period (see S2 File COVID-19 Questionnaire). The answers to the remaining questions were to be delivered using either a multiple-choice scale or standard sliders derived from the Affective Slider. In the case of the latter, subjects rated their level of agreement on a scale ranging from "not at all" to "very much". The questionnaire was administered at the end of the experiment. For the analysis, we included only the data of those subjects who completed the questionnaire.

## Procedure

The online experiment consisted of four main sections: (a) instructions, the consent form, disclaimer, as well as the collection of demographic data (gender, age, education level, country of origin, and country of residence), (b) experimental task, (c) COVID-19 questionnaire, and (d) explanation of the rationale of the study.

During the experimental task, each participant was presented with a sequence of thirty affective stimuli from the OASIS image set [40]. Participants provided their ratings using the Affective Slider located on the right side of the image (Fig 1). Each stimulus remained visible until the submission of both ratings, which had no time limit, as in the experimental tasks of both the tool [26] and the data set [40]. Only when both ratings were provided, subjects could advance to the next image by clicking a separate button. After that, the next stimulus was immediately displayed together with the corresponding Affective Slider.

Once participants completed the experimental task, they were required to complete the COVID-19 questionnaire. Finally, after having submitted the questionnaire, participants were presented with a final page that included the experimental rationale and the researchers' contact information.

## Data analysis

Tests of normality were performed on the data, and subsequently, T-tests were used to identify differences between the affective ratings. All comparative analyses used two-tailed tests and a standard level of significance ($p < .05$). For each comparison, the effect sizes were computed using Cohen's $d$ [42]. A Pearson product-moment correlation coefficient was computed for the subsequent linear correlation analyses. Fourteen participants who reported not being in quarantine were excluded from the analysis.

Finally, we applied machine learning techniques to evaluate the plausibility of predicting participants' personal situation and reported subjective state during the quarantine based on their valence ratings provided during the experiment. To achieve this, we trained a C-Support Vector Classification (SVC) model. Parameter tuning was performed using a grid search algorithm. The model was cross-validated to evaluate its performance based on the F-score. The classification was performed using the Scikit-learn machine learning library [43].

## Results

First, we assessed the linear relationship between the affective ratings of arousal and valence collected in the present experiment and those acquired in the original study [40]. To this end, we computed the mean rating from all the subjects for each of the experimental stimuli and extracted the corresponding mean values from the OASIS data set. The analysis yielded high and significant positive correlation between the mean scores for both arousal ($r(30) = .77$, $p < .001$, see Fig 3A) and valence ($r(30) = .88$, $p < .001$, see Fig 3B).

Second, to test our hypothesis, we evaluated the existence of possible shifts in the affective ratings between the present study and the OASIS for the subsets of neutral and positive stimuli. In the neutral subset, we included all the images whose mean ratings for valence ranged

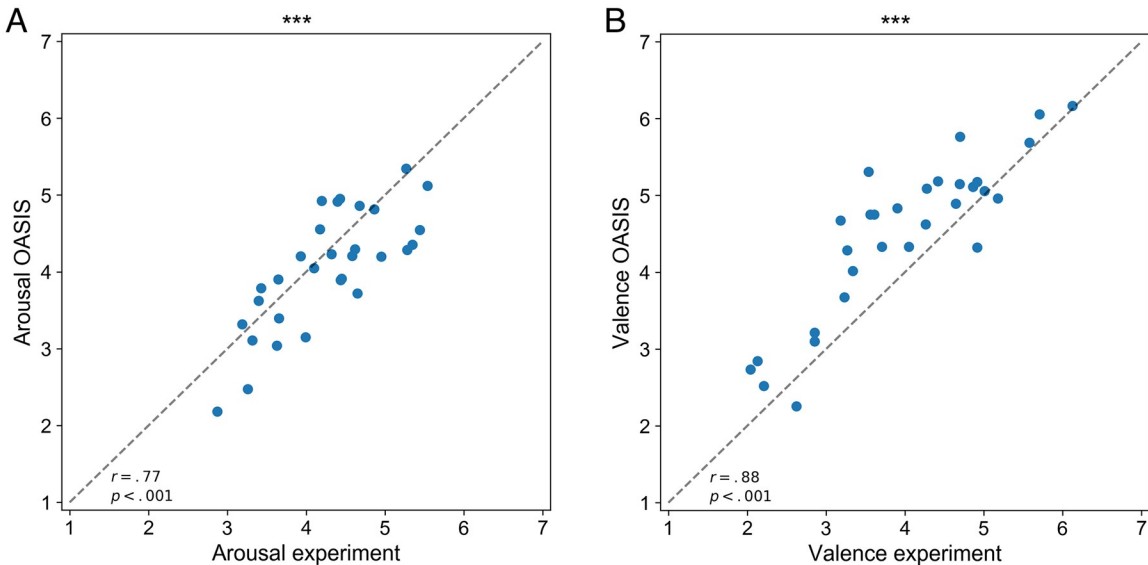

**Fig 3. Linear correlations between the ratings obtained in our study and those from OASIS.** A: Linear correlation between arousal ratings from OASIS (y-axis) and those acquired in the present study (x-axis). B: Linear correlation between valence ratings from OASIS (y-axis) and those acquired in the present study (x-axis). In both graphs, dashed lines represent the identity lines; ***$p$ <.001.

between 3 and 5 ($N = 15$), while in the positive one, those whose mean valence ratings ranged between 5 and 7 ($N = 11$). For these analyses, we computed the mean rating of both arousal and valence from all subjects for each chosen subset. For the neutral stimuli, statistical analyses yielded that, while the mean ratings of arousal for the chosen images did not differ ($t(15) = .61$, $p = .546$), there was a statistically significant negative shift in the ratings of valence ($t(15) = -2.28$, $p = .030$, $d = .859$, see Fig 4).

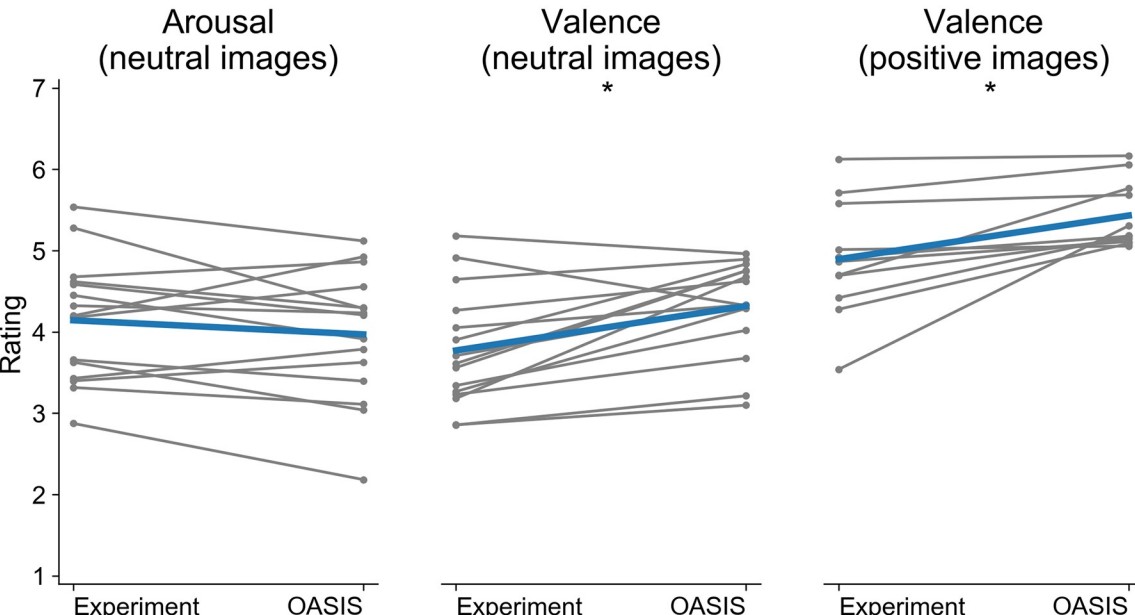

**Fig 4. Shift in the affective ratings for neutral and positive images.** The graphs present the comparison between the ratings of arousal (left) and valence for neutral images (middle) and valence for positive ones (right) obtained in our study with those from the OASIS. In all graphs, the blue lines correspond to the mean, while the individual lines show differences for individual images ($N = 15$); *$p$ <.05.

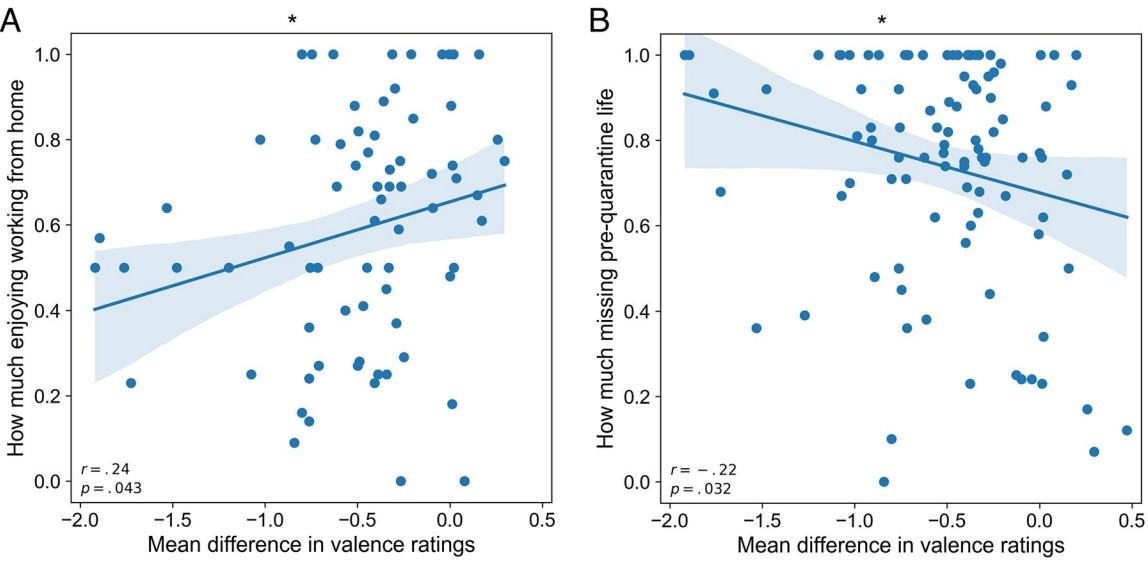

**Fig 5. Correlations between the differences in valence ratings per participant and self-reported situation during the quarantine period.** A: Linear regression between differences in valence ratings and the degree of enjoyment to work from home. B: Linear regression between differences in valence ratings and the degree of missing the "normal" pre-quarantine life. In both graphs, blue lines represent a linear regression fit; $^*p$ <.05.

Similarly, for the positive stimuli, we found no differences in the mean ratings of arousal ($t(11) = 1.313$, $p = .203$). In line with literature, however, we found a statistically significant negative shift in the ratings of valence ($t(11) = -2.148$, $p = .044$, $d = .974$).

Third, we conducted post hoc analyses to assess relationships between the affective ratings of valence and participants' situation during the quarantine period evaluated through the COVID-19 questionnaire. Specifically, we investigated if the mean ratings of valence are related to whether the subjects (a) enjoy working from home, (b) miss the "normal" pre-quarantine life, and (c) live alone. For these analyses, we computed the differences in mean ratings from the present study and the OASIS data set for each participant. The first correlation analysis yielded a significant positive linear relationship between the strength of the enjoyment of working from home and the mean difference in valence ratings ($r(98) = .24$, $p = .043$). In particular, we found that participants who reported enjoying working from home rated the images more positively than those who did not (Fig 5A). Second, our results revealed a significant negative correlation between the degree of missing the "normal" pre-quarantine life and the differences in valence ratings ($r(98) = -.22$, $p = .032$). Hence, participants who missed more to return to the normal life rated images more negatively than those who missed it less (Fig 5B).

We also report a difference in the ratings of valence between those subjects who lived alone and those who lived with their families, partners, or friends ($t(98) = -2.42$, $p = .017$, $d = .611$). Specifically, we found that participants living alone rated the images significantly more negatively (Fig 6).

Fourth, we analyzed the time that participants took to rate each image. To do this, we computed the median rating time for each participant. The D'Agostino-Pearson normality test revealed that the rating times were not normally distributed ($p < 0.001$). Hence, similar to other studies [44], we applied nonparametric statistics for the subsequent analyses of rating times. We found a significant positive correlation between ratings times and both arousal ($r(98) = .32$, $p = .001$) and valence ($r(98) = 0.25$, $p = .012$).

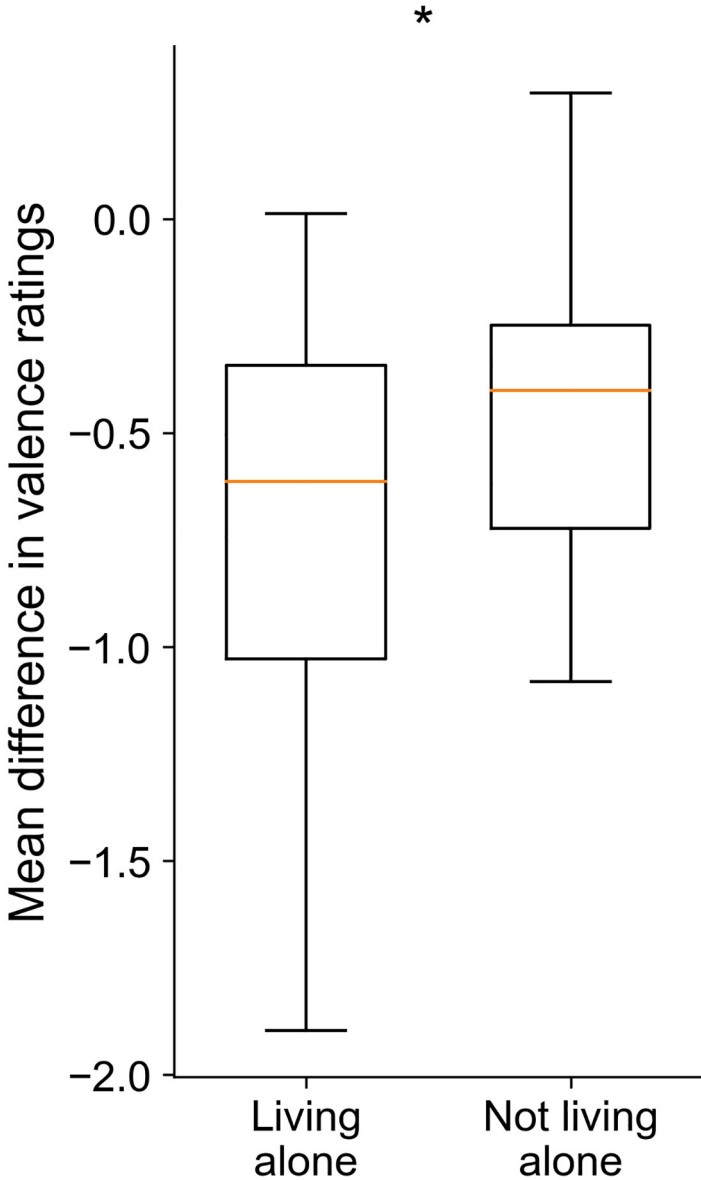

**Fig 6. Differences in valence ratings relative to those of OASIS between participants who, during the quarantine, lived alone and those who did not.** * $p < .05$.

Finally, we applied machine learning techniques to showcase the potential of automatically detecting users who might be at risk of developing mood disorders based on their ratings. To achieve this, we trained an SVC classifier with the valence rating information and the questionnaire's key answers. The proposed method was able to classify between those participants who lived alone and those who lived with other people with a mean accuracy of 84% ($SD = 4$). Additionally, another SVC classifier could determine whether participants missed the pre-quarantine life with an accuracy of 65% ($SD = 4.5$).

## Discussion

In this study, we aimed at assessing the effects of the COVID-19 quarantine on the emotional state of the affected individuals. We predicted that the quarantine restrictions and, in particular,

the lock-down might negatively impact mental health. It has been shown that mood deviations are reflected in the perception of affective stimuli. Hence, to test our hypothesis, we devised an online study whereby volunteers evaluated the arousal and valence of a set of standardized stimuli and compared the acquired scores with those from the original data set. We predicted that the current ratings of valence might be lower than those of OASIS, possibly due to the recruited participants' personal and social situation during the confinement.

Our results revealed that individuals who, during the experiment, were undergoing the quarantine due to COVID-19 rated neutral stimuli as significantly less pleasant when compared to the subjects who evaluated the same images during a non-quarantine period. We propose that the reported shifts in the valence ratings might be further indicative of a more general negative affective state caused by the quarantine. Indeed, we find evidence about negative changes in perception, as measured through self-reported valence ratings of visual stimuli in people with depression compared to healthy controls [30].

Based on the acquired data, we further observed a significant effect of some of the critical aspects of our sample's personal and working situation during the self-isolation period on the reported ratings. Our results revealed a positive relationship between how much the subjects enjoyed working from home during confinement and the affective ratings. On the one hand, this finding is consistent with the literature, which demonstrates that unemployed people tend to report higher episodic sadness levels than employed people [45]. On the other hand, this result might indirectly represent the effect of a decreased in-person social interaction that many jobs entail, provided that social interaction positively impacts psychological well-being [46].

The experience of missing regular life before the quarantine also yielded a significant effect on the negativity of the emotive ratings. We found that those participants who missed it more also experienced more substantial negative shifts in the affective assessments of the stimuli than those who missed it less. As previously demonstrated [6–8], we speculate that this relationship might be directly indicative of the lowered mood stemming from the negative perception of the current situation and the desire for the social distancing measures and self-quarantine to terminate. This, in turn, may be related to an increased need for both social interaction and freedom.

Furthermore, our results revealed that the ratings of valence differed depending on the participants' social living situation. Specifically, those individuals who lived alone provided more negative ratings than those living with other people. This might suggest that increased social isolation and reduced social interaction in individuals who undergo the quarantine while living alone more negatively impact their perception and, possibly, mood. Indeed, ample scientific evidence demonstrates that social isolation can result in lowered mood and depression and induce many other adverse effects on health [47]. These effects can range from mental disorders such as depression or anxiety [48–50] to cardiovascular diseases [51, 52]. Moreover, loneliness can have detrimental effects on health through several mechanisms, including health behaviors, cardiovascular activation, cortisol levels, and sleep [53]. Although social isolation and loneliness are prevalent in a large proportion of the general population, affecting both younger [54] and older [55, 56] adults, these conditions can be exacerbated or become even more strict under exceptional circumstances that force a decrease in social contact. In the case of the COVID-19 pandemic, several studies also point out a significant psychological impact, including symptoms that correspond to those found in social isolation [57–59].

The above-discussed findings converge to suggest that the mitigation strategies employed to prevent the spread of the COVID-19 pandemic are negatively impacting the emotional state of the affected individuals, which is reflected by negative shifts in the ratings of the affective stimuli. Furthermore, this pernicious effect is exacerbated by personal circumstances related to

working conditions and social isolation, which, in the long term, might result in an increased prevalence of mental health conditions such as depression or post-traumatic stress disorder [60]. Importantly, in the present paradigm, we focused primarily on the evaluation of neutral and positive stimuli. According to literature [30], however, one could expect that quarantine-induced disorders of mood might also result in shifts in the negative stimuli—the hypothesis we are currently addressing in a follow-up study.

It is worth noting that our data presented variability in the relationships between the mean difference in valence ratings and both the enjoyment of working from home and the feeling of missing life from before the quarantine. This may be explained by the interaction of additional factors that were not captured by the present experiment but might have impacted the participants' emotional state. For example, personality traits might play an essential role in the ways individual participants are affected by social isolation and how they cope with it [61–63]. Furthermore, the intensity of the enforced quarantine measures was not the same for all participants, resulting variation in self-isolation. Future studies should address these limitations by controlling for additional, possibly confounding factors. Moreover, the participant sample used in this study comes from a variety of European countries. This sampling approach was intentionally chosen to cover a set of regions with comparable cultures as well as quarantine and self-isolation measures. It is possible, however, that the underlying diversity of the sample could have introduced heterogeneity in the data, which could impact the generalizability of our findings. This limitation shall be addressed in future studies by focusing the collection of data from a smaller subset of countries to further ensure the commonality of demographic aspects that could better represent the mental health of the sampled population.

On the one hand, the outcome of this study highlights the impact of the COVID-19-induced quarantine on the affective states, thus emphasizing the need for continuous monitoring of the psychological health and well-being of the general population. Since the psychological effects of isolation might have long-term consequences, the identification of individuals at risk and carrying out interventions to mitigate the reported negative impact might be necessary not only during but also post-quarantine. On the other hand, the hereby proposed method for diagnosing the affective changes through subjective ratings of emotive stimuli may already be of use to the healthcare system. Specifically, the current findings, as well as the reported machine learning techniques, could be translated into clinical practice by using techniques such as in-person visits and digital technology in the form of smartphone apps. The former could provide a unique opportunity of combining multidimensional scales including, for instance, brain scanning (e.g., functional Magnetic Resonance Imaging) genomic measurements, observer-rated neurocognitive evaluations (e.g., HDRS), patient self-reports (e.g., BDI), medical record reviews, as well as implicit measures such as the affective evaluations used in our study. From the academic and medical perspectives, such a compound diagnosis could contribute to fundamental advances in understanding neuropsychological conditions. However, there is a need for easy to apply and low-cost solutions for diagnostics, monitoring, and treatment. Hence, the implicit assessment validated in our study can allow continuous monitoring of the effective ratings as the proxy of the affective states allowing for a prediction of the personal situation based on the obtained ratings. Such software could promote at-home remote diagnostics and monitoring of at-risk patients continuously, at a low cost, and with a further benefit of preventing possible response biases [23–25]. We have successfully deployed such an approach in the domain of stroke rehabilitation. We have successfully deployed such an approach in the domain of stroke rehabilitation [64, 65]. To this end, in future studies, we shall more systematically investigate the specific factors that may influence the participants' affective ratings, including personality type, as well as other symptoms that might indicate

abnormal psychological states, such as insomnia. Moreover, we will further validate the statistical relationship between the proposed implicit measure of the affective states and standard tools used to evaluate the mood, such as BDI [66] or PHQ-9 [67].

The efficient diagnosis, monitoring, and treatment of a neuropsychiatric illness are becoming increasingly important because its burden exceeds that of cardiovascular disease and cancer [68] and it is estimated that about 25% of individuals will suffer neurological or mental disorders at some point in their lives. However, due to several factors, including the lack of trained healthcare professionals, pervasive underdiagnosis, and stigma, only 0.2% will be able to receive the necessary treatment [69]. Hence, key current challenges include the improvement of the efficacy of the diagnosis of psychological disturbances and overcoming known limitations of current clinical scales [16–20, 22] together with accurately capturing symptoms and patient specific concerns [70]. To this end, we propose that an optimal evaluation strategy may comprise explicit, observer-rated and self-reported evaluation tools combined with implicit physiological and behavioral monitoring using biometric sensing, such as the proposed affective rating methods and associated tools [71].

Importantly, at the current stage, the proposed classification algorithms serve rather as proof of the potential to automatically classify well-being [72]. Future work will address this limitation by further improving the model. Those improvements will imply additional training of the classifier and the inclusion of supplementary variables that might affect participants' mental state, such as personality traits and biometrics.

Additionally, the present findings support the notion that the results from online studies carried out during the quarantine period that rely on the assessment of affective ratings or similar, might be significantly affected. Hence, this impact should be considered in the analyses and the interpretation of the acquired results.

Taken together, the present report presents a significant and timely finding which sheds light on the current quarantine's impact beyond the experience of the individuals who undergo it. In line with other studies [5, 11–13] our results confirm that individuals undergoing current mass quarantine can experience adverse psychological effects and be at risk of anxiety, mood dysregulations, and depression, which, in the long term, may lead to post-traumatic stress disorder and affect overall wellbeing [6–8]. Indeed, according to previous studies, the measures that are commonly undertaken to mitigate pandemics, including stay-at-home rules and social distancing may have drastic consequences. For instance, people can experience intense fear and anger leading to severe consequences at cognitive and behavioral levels, culminating in civil conflict and legal procedures [6] as well as suicide [9, 10]. In addition, the long-term impact of this change in wellbeing is currently not understood and deserves further study. The results presented in this report highlight the need to explore possible impacts of the COVID-19 pandemic and its effects on psychological wellbeing and mental health. To this aim, more studies need to be conducted to systematically investigate the interventions that may be deployed by both the healthcare system and individuals undergoing quarantine to mitigate the adverse psychological effects.

## Supporting information

**S1 File. Selection of images used in this study from the OASIS data set.**
(PDF)

**S2 File. The COVID-19 questionnaire used in the study.**
(PDF)

## Author Contributions

**Conceptualization:** Klaudia Grechuta.

**Data curation:** Héctor López-Carral.

**Formal analysis:** Héctor López-Carral.

**Funding acquisition:** Paul F. M. J. Verschure.

**Investigation:** Héctor López-Carral.

**Methodology:** Héctor López-Carral, Klaudia Grechuta.

**Supervision:** Paul F. M. J. Verschure.

**Validation:** Paul F. M. J. Verschure.

**Writing – original draft:** Héctor López-Carral, Klaudia Grechuta.

**Writing – review & editing:** Klaudia Grechuta, Paul F. M. J. Verschure.

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
