## [Decision Letter · Decision Letter 0]

16 Jul 2020

PONE-D-20-15454

Subjective ratings of emotive stimuli predict the impact of the COVID-19 quarantine on affective states

PLOS ONE

Dear Dr. Carral,

Thank you for submitting your manuscript to PLOS ONE. After careful consideration, we feel that it has merit but does not fully meet PLOS ONE’s publication criteria as it currently stands. Therefore, we invite you to submit a revised version of the manuscript that addresses the points raised during the review process.

We look forward to receiving your revised manuscript.

Kind regards,

Stephan Doering, M.D.

Academic Editor

PLOS ONE

2.Thank you for including your ethics statement: 

"The study was approved by the ethics committee of the researchers' institution.

Participants consented to participate and were informed of their rights before starting

the experiment. No personal data of the participants were recorded".   

Once you have amended this statement in the Methods section of the manuscript, please add the same text to the “Ethics Statement” field of the submission form (via “Edit Submission”).

4.Thank you for stating the following in the Competing Interests section:

[I have read the journal's policy and the authors of this manuscript have the following competing interests: PFMJV is the founder and interim CEO of Eodyne S L, which aims at bringing scientifically validated neurorehabilitation technology to society. The rest of the authors have nothing to disclose.].

5.   We note that Figures supporting 1 and Figure 1 in your submission contain copyrighted images. All PLOS content is published under the Creative Commons Attribution License (CC BY 4.0), which means that the manuscript, images, and Supporting Information files will be freely available online, and any third party is permitted to access, download, copy, distribute, and use these materials in any way, even commercially, with proper attribution. For more information, see our copyright guidelines: http://journals.plos.org/plosone/s/licenses-and-copyright.

a)        You may seek permission from the original copyright holder of Figure(s) [#] to publish the content specifically under the CC BY 4.0 license.

Reviewers' comments:

Reviewer's Responses to Questions

**Comments to the Author**

1. Is the manuscript technically sound, and do the data support the conclusions?

Reviewer #1: Partly

Reviewer #2: Yes

2. Has the statistical analysis been performed appropriately and rigorously? 

Reviewer #1: Yes

Reviewer #2: Yes

3. Have the authors made all data underlying the findings in their manuscript fully available?

Reviewer #1: Yes

Reviewer #2: Yes

4. Is the manuscript presented in an intelligible fashion and written in standard English?

Reviewer #1: Yes

Reviewer #2: Yes

5. Review Comments to the Author

Reviewer #1: PONE-D-20-15454

Effects of COVID-19 quarantine on affective states

Thank you for the opportunity to review this novel approach to assessing affective states in people impacted by quarantine. There are 2 principle concerns wit this manuscript as currently presented.

1. There are some grammatical suggestions to improve the readability of the manuscript

2. The r values have not been interpreted at all the strength of the relationships not discussed

Specific recommendations are outlined below:

L4-7. I recommend updating data to the most recent figures at the time of publication

L8. Suggest; …public health authorities have employed…

L13. Previous outbreaks of what? Are you referring to SARS/MERS?

L16. The statement about a growing body of evidence is not substantiated with the provided reference. Please ensure relevant references are provided to support this claim.

L26. This statement is supported by 2 references, both pertaining to a single depression rating scale, one of which is far from current, published in 1993. If this statement were true why is the HDRS still widely used?

L61. Was the required sample size reached?

L69. Why was it relevant to blind participants to the study purpose and how was this done? This suggests some degree of deception, or was this address in part D of the online experiment?

L183. Why was median and not mean time to rate each image used?

L220. Statistically these correlations are significant but the scatterplots (Figure 5) show substantial variability hence the strength of the correlations are weak at best. Please comment on this in the discussion. This comment also applies to the strength of the correlations based on SVC which are only weak to moderate (L184).

L265-269. Other than the response bias issues, what is the significance of reporting personal situations? I am not sure these are ‘robust’ with rating at 65-84% accuracy. These are far lower than that reported, for example, for sentiment analysis of social media posts, but might reflect the novel application of ML to this topic. On the other hand, it might indicate the need for more training of the algorithm.

L269-275. This section focussed solely on the implementation of the technology, not the psychological health of participants. I feel more emphasis on the interpretation of the findings is needed, rather than discussing future application of the model.

Reviewer #2: This study provides valuable insights regarding emotive perception among individuals under quarantine in COVID-19 pandemic. The findings of this study may inform future research and policymaking on mental health, especially for people who are more likely to have impaired affective states. However, this study may be subjected to a methodological concern related to sampling and comparative analyses, which should be considered prior to communicating this research with a broader audience.

As a small sample was drawn from a diverse online population from 19 countries, it is likely that their mental health may not represent the populations they belong to. Furthermore, demographic and psychosocial factors in the study sample may be heterogenous in nature, which may further affect the generalizability of the findings. Proper rationale for this sampling approach should be presented in the methods section and associated limitations should be discussed in the discussion section of the article.

Another perspective on the use of the evidence revealed in this study is how mental health practitioners and policymakers can translate the findings into clinical practice and mental health policymaking. The authors may wish to draw some inferences on how the altered emotive perceptions may result in short- and long-term mental health impacts, how some individuals are more vulnerable than others, and how potential strategies can be adopted to mitigate such mental health challenges during this and future pandemics.

6. PLOS authors have the option to publish the peer review history of their article (what does this mean?). If published, this will include your full peer review and any attached files.

Reviewer #1: No

Reviewer #2: No

---

## [Author Response · Author response to Decision Letter 0]

24 Jul 2020

Editors

R1. We thank the Editor for this advice. As suggested, we have carefully examined the journal’s style requirements, including those for file naming, and adjusted the files accordingly. We believe that the current version of the manuscript, as well as the additional materials, fully comply with all the requirements.

R2. We have amended our ethics statement specifying the name of the ethics committee that approved the reported experiment. Thank you for raising this issue.

R3. Thank you for raising this point. We would like to confirm our commitment to making all data available in a repository when the manuscript is accepted.

R4. We confirm that the declared competing interests do not alter our adherence to PLOS ONE policies on sharing data as materials. As such, we have added a clarifying statement in our cover letter:

“The authors of this manuscript have the following competing interests: PFMJV is the founder and interim CEO of Eodyne S.L., which aims at bringing scientifically validated neurorehabilitation technology to society. This does not alter our adherence to PLOS ONE’s policies on sharing data and materials. The rest of the authors have nothing to disclose.”

5. We note that Figures supporting 1 and Figure 1 in your submission contain copyrighted images. All PLOS content is published under the Creative Commons Attribution License (CC BY 4.0), which means that the manuscript, images, and Supporting Information files will be freely available online, and any third party is permitted to access, download, copy, distribute, and use these materials in any way, even commercially, with proper attribution.

R5. We thank the Editor for raising this point. We would like to clarify that all the images used both in Figure 1 and Figures supporting 1 come from the OASIS dataset (Kurdi et al., 2017), which we used as the benchmark for the emotive rating evaluation. The authors of this dataset explicitly indicate that all the images “can be downloaded, used, and modified free of charge for research purposes” (please see https://pixabay.com/service/license/). Similar, in the case of the image used in Figure 1, the source states that its license allows “free for commercial and noncommercial use across print and digital” and that “attribution is not required” (please see

https://pixabay.com/photos/man-human-person-alone-being-alone-396299/).

This confirms that the images in question are not subject to copyright restrictions; hence they can be published under the Creative Commons Attribution License (CC BY 4.0). 

Reviewer #1

Thank you for the opportunity to review this novel approach to assessing affective states in people impacted by quarantine. There are 2 principal concerns with this manuscript as currently presented.

Dear Reviewer, we thank you for the encouraging feedback regarding our work. Your suggestions have contributed to the improvement of our manuscript. Below, we provide detailed responses to all your comments.

1. There are some grammatical suggestions to improve the readability of the manuscript 

R1. We thank the Reviewer for this valuable comment. To improve readability, we have rephrased long, complex, or unclear sentences, corrected the typographical errors, and the manuscript was run through a professional grammar and spell check platform for scientific writing. Subsequently, it was revised by a native speaker of English with a background in medicine and neuroscience. We believe that by addressing this point, we have significantly increased the legibility of the manuscript.

2. The r values have not been interpreted at all the strength of the relationships not discussed

R2. We thank the Reviewer for raising this point. We agree with the Reviewer that the manuscript was lacking an explicit interpretation of the r values. Since the concern is very much related to comment #11, we have addressed both in response R11. Please, see below.

Specific recommendations are outlined below:

3. L4-7. I recommend updating data to the most recent figures at the time of publication

R3. This is a fair point, thank you. We have updated all the reported data as follows (Lines: 4-7):

“By mid-March 2020, a total of 200,000 confirmed cases (Johns Hopkins, 2020) have been reported worldwide, showing an exponential increase with the current number of identified cases exceeding 14 million, whereby Spain, Italy, and the United Kingdom are the most-affected European nations.”

4. L8. Suggest; …public health authorities have employed…

R4. We have followed the suggestion of the Reviewer and changed the text accordingly. 

5. L13. Previous outbreaks of what? Are you referring to SARS/MERS?

R5. Thank you for noticing this. What we actually mean are previous applications of quarantine, rather than the outbreaks. We have clarified the sentence as follows (Lines: 13-15):

“Indeed, prolonged widespread lock-down and limiting social contact has resulted in post-traumatic stress disorder, depression, anxiety, mood dysregulations, and anxiety-induced insomnia during previous periods of quarantine (Miles et al., 2015, Brooks et al., 2020, Hossain et al., 2020).” 

6. L16. The statement about a growing body of evidence is not substantiated with the provided reference. Please ensure relevant references are provided to support this claim.

R6. Thank you for noticing this. To further support our statement, we have included the following relevant references:

Holmes, E. A., O’Connor, R. C., Perry, V. H., Tracey, I., Wessely, S., Arseneault, L., Bullmore, E. (2020). Multidisciplinary research priorities for the COVID-19 pandemic: a call for action for mental health science. The Lancet Psychiatry. Elsevier Ltd. https://doi.org/10.1016/S2215-0366(20)30168-1

Rajkumar, R. P. (2020). COVID-19 and mental health: A review of the existing literature. Asian Journal of Psychiatry, 52. https://doi.org/10.1016/j.ajp.2020.102066

Torales, J., O’Higgins, M., Castaldelli-Maia, J. M., & Ventriglio, A. (2020). The outbreak of COVID-19 coronavirus and its impact on global mental health. International Journal of Social Psychiatry, 66(4), 317–320. https://doi.org/10.1177/0020764020915212

7. L26. This statement is supported by 2 references, both pertaining to a single depression rating scale, one of which is far from current, published in 1993. If this statement were true why is the HDRS still widely used?

R7. The reason why we are focusing on the criticism of the HDRS (Hamilton 1960), specifically, was twofold. First, HDRS has constituted the most common observer-rated instrument to measure depression severity, its changes over time, and the efficacy of treatment for over 60 years. Second, it has been regarded as the gold standard in clinical trials (Wiliams 2001, Bech 2009, Bagby et al., 2004, Gibbons et al., 1993, Stefanis et al., 2002, Gullion & Rush 1998). As described in the manuscript, despite the extensive use of HDRS, the scale seems to present several limitations worth noting. Below, we provide the specific criticisms discussed in the two aforementioned references, among others, and address the Reviewer’s comment related to the extensive use of the scale independent of its reported limitations.

The first reference (Bagby et al., 2004), which we included to support our statement, systematically examined 70 articles that aimed to explicitly evaluate the psychometric properties of the HDRS, conceptual issues related to its development, continued use, and shortcomings. The studies included in that review were published between January 1980 and May 2003. The authors found that, although the internal reliability at the item level was mostly satisfactory, a significant number of scale items were, in fact, poorly contributing to the measurement of depression severity, and many items presented low inter-rater and test-retest reliability (Bagby et al., 2004). The authors argued that while the convergent validity and discriminant validity were adequate, content validity was quite unsatisfactory. Furthermore, the scale was designed as multidimensional, resulting in weak replication across samples. Finally, the analysis yielded that the response format was biased such that certain items contributed more to the total score than others. Indeed, according to the psychometric model (Shapiro 1951), each of the scale items should display identical clinical weight (Fava & Belaise, 2005). A similar criticism to those of Bagby et al. was reported by Zimmerman et al., (2004). In particular, the authors emphasized that the differential item weight in HDRS, whereby some items contribute more to the total score than others, is the most critical limitation of the scale. To further support our statement, we have included this reference within the text (Line: 28). At the end of their reviews, both Bagby et al., (2004) and Zimmerman et al., (2004) concluded that, because of the lack of factorial and content validity, HDRS is a flawed measure and suggested replacing it with a novel, more sensitive gold standard for the assessment of depression. 

The second reference (Gibbons et al., 1993) emphasizes the necessity to examine and reassess the robustness of clinical ratings in general, including the HDRS, to prevent an unreliable diagnosis of psychological phenomena and weakened validity of the studies that use such scales. The authors approach their analysis by investigating the flaws of HDRS in the context of two fundamental principles of psychometric assessment, including (1) defining a syndrome and scaling its severity, and (2) considering the issue of multidimensionality of the scale. As discussed above, HDRS presents flaws regarding both parameters. 

Additionally to the two citations included in the initial version of the manuscript, the HDRS scale was further criticized by others as a nonobjective measure of depression severity since it does not correlate with other clinical assessments and it does not permit a definition of a unidimensional depressive state (Bech & Allerup 1981, Maier, Philipp, & Gerken 1985, Fried & Nesse 2015, among others). We have included these references to the body of the text (Line: 28). Moreover, to explicitly address this point in more depth within the manuscript, we have added the following lines to the Introduction section (Lines: 28-32):

“ (...) For instance, the Hamilton Depression Rating Scale (HDRS, Gibbons et al., 1993), which has been considered a gold standard in clinical practice as well as clinical trials, was widely criticized for its subjectivity as well as the multidimensional structure, which varies across studies hence preventing replication across samples as well as poor factorial and content validity (Bagby et al., 2004, Zimmerman et al., 2004, Beth & Allerup 1981, Maier, Philipp, & Gerken 1985)”.

To answer the final question, we propose that, despite its limitations, including subjectivity, HDSR is still used in clinical practice first because of its long tradition and second because of the lack of well established and robust alternative measures. Indeed, other standardised, yet also criticized, self-reported depression scales such as the Beck Depression Inventory (BDI, Beck et al., 1988) or the Zung Depression Rating Scale are also widely used in the clinic and in academia. Most research related to depression is still grounded in just those few scales, including HRSD (and, specifically, its 20 variations) and BDI, none of which is shown to be sufficiently robust (Santor et al., 2009). Since our understanding of depression depends mostly on the quality and accuracy of the diagnosis, monitoring, and treatment, we argue that there is a need for novel depression assessment tools which would allow replication across samples while accommodating the heterogeneity and diversity of depressive disorders, including major depressive disorder (MDD, Kessler et al., 2003). Such an approach will not only facilitate our understanding of the specific causes of depression but also better inform clinical decision-making. Possibly, a complete solution would require a combination of explicit and implicit assessment and monitoring tools delivered continuously to not only diagnose developing mental disorders but also prevent them (Sayers 2001). Such tools should include an assessment of symptomatology and the wellbeing of patients (Demyttenaere et al., 2020). This is even more relevant now that the burden of neuropsychiatric illness exceeds that of cardiovascular disease or cancer (Vigo et al., 2016). We have addressed this analysis and its generalization in the following way in the Discussion section (Lines: 318-331):

“The efficient diagnosis, monitoring, and treatment of neuropsychiatric illness are becoming increasingly important because its burden exceeds that of cardiovascular disease and cancer (Vigo et al., 2016) and it is estimated that about 25% of individuals will suffer neurological or mental disorders at some point in their lives. However, due to several factors, including the lack of trained healthcare professionals, pervasive underdiagnosis, and stigma, only 0.2% will be able to receive necessary treatment (Sayers 2001). Hence, key current challenges include the improvement of the efficacy of the diagnosis of psychological disturbances and overcoming known limitations of current clinical scales (Bagby et al., 2004, Zimmerman et al., 2004, Beth & Allerup 1981, Maier, Philipp, & Gerken 1985) together with accurately capturing symptoms and patient specific concerns (Demyttenaere et al., 2020). To this end, we propose that an optimal evaluation strategy may comprise explicit, observer-rated and self-reported evaluation tools combined with implicit physiological and behavioral monitoring using biometric sensing, such as the proposed affective rating methods and associated tools (Reinertsen & Clifford 2018).”

8. L61. Was the required sample size reached?

R8. We thank the reviewer for raising this point. As estimated by the G*Power software, the required sample size equaled 110 participants, while the total number of subjects who participated in our study was N= 112. We have specified that in the Participants subsection of the Methods section as follows (Lines: 65-66):

“The sample size of N= 110 was determined a priori using the G*Power software version 3.1 (Kiel, Germany) based on α= 0.05, power of 80% and medium effect size (0.5).”

9. L69. Why was it relevant to blind participants to the study purpose and how was this done? This suggests some degree of deception, or was this address in part D of the online experiment?

R9. We designed the experiment such that the participants were blind to the purpose of the study. Specifically, they were not aware that the study's objective was to investigate possible adverse effects of the COVID-19 pandemic on the affective ratings until the end of the session. We wanted to prevent this information to bias the ratings of the emotive stimuli. To this end, the experiment consisted of four main sections including

instructions, the consent form, disclaimer, and the collection of demographic data,

experimental task,

COVID-19 questionnaire, and

explanation of the rationale of the study.

With THIS design, the subjects provided the affective ratings without being aware that they may be informative of their mood/affective state. In section 4, however, after completing the whole experiment, we provided the study's full rationale and explained our hypothesis. To address this point, we have included the following description to the Participants subsection of the Methods section (Lines: 80-82):

“Specifically, until the end of the session, subjects did not know the study’s objective, which could bias their responses. However, they were informed about it at the end of the trial.”

10. L183. Why was median and not mean time to rate each image used?

R10. Thank you for raising this point. We used the median time instead of the mean time to accurately quantify the time that the users took to rate each image. The median is a common statistic used to analyse reaction times, favored over the mean, as long as the number of trials is the same in all cases (as it is in our experiment). This compensates for the skewed distribution due to intermittent long reaction times (Whelan, 2008). Indeed, our data reflects a similar trend. In particular, the D’Agostino-Pearson normality test revealed that the reported reaction times were not normally distributed (p < 0.001). To clarify this, we have included the following information in the Results section (Lines: 195-198):

“The D’Agostino-Pearson normality test revealed that the rating times were not normally distributed (p < 0.001). Hence, similar to other studies (Whelan, 2008), we applied nonparametric statistics for the subsequent analyses of rating times.”

11. L220. Statistically these correlations are significant but the scatterplots (Figure 5) show substantial variability hence the strength of the correlations are weak at best. Please comment on this in the discussion. This comment also applies to the strength of the correlations based on SVC which are only weak to moderate (L184).

R11. This is a valuable comment, thank you. To address this point, we have included the following paragraph in the Discussion section (Lines: 271-280):

“It is worth noting that our data presented variability in the relationships between the mean difference in valence ratings and both the enjoyment of working from home and the feeling of missing life from before the quarantine. This may be explained by the interaction of additional factors that were not captured by the present experiment but might have impacted the participants' emotional state. For example, personality traits might play an essential role in the ways individual participants are affected by social isolation and how they cope with it (Taylor et al., 1969; Kong et al., 2014; Zelenski et al., 2013). Furthermore, the intensity of the enforced quarantine measures was not the same for all participants, resulting in variation in self-isolation. Future studies should address these limitations by controlling for additional, possibly confounding factors.”

12. L265-269. Other than the response bias issues, what is the significance of reporting personal situations? I am not sure these are ‘robust’ with ratings at 65-84% accuracy. These are far lower than that reported, for example, for sentiment analysis of social media posts, but might reflect the novel application of ML to this topic. On the other hand, it might indicate the need for more training of the algorithm.

R12. This is an important point. Indeed, we report different aspects related to the personal situation of the participants since, as revealed in previous studies, they comprise significant indicators of affective states during quarantines. Specifically, as we describe in the Introduction section as well as in the Discussion, certain situations, such as physical interaction with others, might significantly affect participants’ mood (Lines: 10-12):

“Mandatory mass quarantine restrictions, which include social distancing, stay-at-home rules, and limiting work-related travel outside the home (Rothstein et al., 2003), might impact both physical and mental health of the affected individuals (Nobles et al., 2020).”

We agree that, ideally, the accuracy of the proposed classification algorithms should have lower variability. However, at the current stage, it rather provides an indication of the robustness of the model we are currently improving (Lines: 332-337):

“Importantly, at the current stage, the proposed classification algorithms serve rather as proof of the potential to automatically classify well-being (Lipton et al., 2014). Future work will address this limitation by further improving the model. Those improvements will imply additional training of the classifier and the inclusion of supplementary variables that might affect participants’ mental state, such as personality traits and biometrics.”

13. L269-275. This section focussed solely on the implementation of the technology, not the psychological health of participants. I feel more emphasis on the interpretation of the findings is needed, rather than discussing future application of the model.

R13. We thank the Reviewer for raising this point. We agree that the discussion related to the participants' psychological health was not thorough enough in the initial version of the manuscript. To address this point, we have included the following paragraph in the Discussion section (Lines: 342-356):

“(...) Taken together, the present report presents a significant and timely finding which sheds light on the current quarantine's impact beyond the experience of the individuals who undergo it. In line with other studies (Nobles et al., 2020, Holmes et al., 2020, Rajkumar et al., 2020, Torales et al., 2020) our results confirm that individuals undergoing current mass quarantine can experience adverse psychological effects and be at risk of anxiety, mood dysregulations, and depression, which, in the long term, may lead to post-traumatic stress disorder and affect overall wellbeing (Miles et al., 2015, Brooks et al., 2020, Hossain et al., 2020). Indeed, according to previous studies, the measures that are commonly undertaken to mitigate pandemics, including stay-at-home rules and social distancing may have drastic consequences. For instance, people can experience intense fear and anger leading to severe consequences at cognitive and behavioral levels, culminating in civil conflict and legal procedures (Miles et al., 2015) as well as suicide (Barbisch et al., 2015, Rubin et al., 2020). In addition, the long-term impact of this change in wellbeing is currently not understood and deserves further study. The results presented in this report highlight the need to explore possible impacts of the COVID-19 pandemic and its effects on psychological wellbeing and mental health. To this aim, more studies need to be conducted to systematically investigate the interventions that may be deployed by both the healthcare system and individuals undergoing quarantine to mitigate the adverse psychological effects.”

Reviewer #2

This study provides valuable insights regarding emotive perception among individuals under quarantine in COVID-19 pandemic. The findings of this study may inform future research and policymaking on mental health, especially for people who are more likely to have impaired affective states. However, this study may be subjected to a methodological concern related to sampling and comparative analyses, which should be considered prior to communicating this research with a broader audience.

1. As a small sample was drawn from a diverse online population from 19 countries, it is likely that their mental health may not represent the populations they belong to. Furthermore, demographic and psychosocial factors in the study sample may be heterogeneous in nature, which may further affect the generalizability of the findings. Proper rationale for this sampling approach should be presented in the methods section and associated limitations should be discussed in the discussion section of the article.

R1. We thank the Reviewer for raising this point. As correctly noted by the Reviewer, our sample comes from a variety of countries. Importantly, however, the vast majority of the participants originate from a relatively small subset of European countries, including Spain (53.57%), Italy (16.07%), Poland (8.04%), and the United Kingdom (5.36%). We consider that this does not impair the generalizability of our results due to cultural similarities that reduce the possible heterogeneity of the data (Gupta et al., 2002). The rationale behind our sampling approach was to include European subjects whose countries apply similar measures to mitigate the spread of the virus [ref]. To explicitly address this issue in our manuscript, we have extended the Methods section (Participants subsection) by including the following (Lines: 71-77):

“This sampling approach was chosen to cover a range of countries that were similarly impacted by self-isolation measures. In particular, for the analyses, we included only those participants who were actively undergoing quarantine. Thus, all participants were uniform in their cultural traits (Gupta et al., 2002) and quarantine measures, including social isolation and distancing, the banning of social events and gatherings, the closure of schools, offices, and other facilities, and travel restrictions (Conti, 2020; Shah et al., 2020).”

Furthermore, we have added the following paragraph in the Discussion section (Lines: 281-289):

“(...) Moreover, the participant sample used in this study comes from a variety of European countries. This sampling approach was intentionally chosen to cover a set of regions with comparable cultures as well as quarantine and self-isolation measures. It is possible, however, that the underlying diversity of the sample could have introduced heterogeneity in the data, which could impact the generalizability of our findings. This limitation shall be addressed in future studies by focusing the collection of data from a smaller subset of countries to further ensure the commonality of demographic aspects that could better represent the mental health of the sampled population.”

2. Another perspective on the use of the evidence revealed in this study is how mental health practitioners and policymakers can translate the findings into clinical practice and mental health policymaking. The authors may wish to draw some inferences on how the altered emotive perceptions may result in short- and long-term mental health impacts, how some individuals are more vulnerable than others, and how potential strategies can be adopted to mitigate such mental health challenges during this and future pandemics.

R2. We thank the reviewer for this comment. To address the issues of translating the proposed technology into the clinical practice and mental health policymaking, we included two paragraphs in the Discussion section. First, we added the following discussion (Lines: 296-311):

“(...) On the other hand, the hereby proposed method for diagnosing the affective changes through subjective ratings of emotive stimuli may already be of use to the healthcare system. Specifically, the current findings, as well as the reported machine learning techniques, could be translated into clinical practice by using techniques such as in-person visits and digital technology in the form of smartphone apps. The former could provide a unique opportunity of combining multidimensional scales including, for instance, brain scanning (e.g., functional Magnetic Resonance Imaging) genomic measurements, observer-rated neurocognitive evaluations (e.g., HDRS), patient self-reports (e.g., BDI), medical record reviews, as well as implicit measures such as the affective evaluations used in our study. From the academic and medical perspectives, such a compound diagnosis could contribute to fundamental advances in understanding neuropsychological conditions. However, there is a need for easy to apply and low-cost solutions for diagnostics, monitoring, and treatment. Hence, the implicit assessment validated in our study can allow continuous monitoring of the effective ratings as the proxy of the affective states allowing for a prediction of the personal situation based on the obtained ratings. Such software could promote at-home remote diagnostics and monitoring of at-risk patients continuously, at a low cost, and with a further benefit of preventing possible response biases (Braun 2001, Paulhus 2002, Paulhus 2017). We have successfully deployed such an approach in the domain of stroke rehabilitation. We have successfully deployed such an approach in the domain of stroke rehabilitation (Ballester at al., 2015, Grechuta et al., 2020).” 

Furthermore, we have included a brief discussion related to the impact of the burden of neuropsychiatric illnesses, including depression and the necessity to rethink current assessment tools provided their limitations. Here, we also comment on the short- and long-term effects of mental health alteration due to COVID-19, as well as propose possible strategies that can be adopted to mitigate such mental health challenges during this and future pandemics. (Lines: 318-331):

“The efficient diagnosis, monitoring, and treatment of a neuropsychiatric illness is becoming increasingly important because its burden exceeds that of cardiovascular disease and cancer (Vigo et al., 2016) and it is estimated that about 25% of individuals will suffer neurological or mental disorders at some point in their lives. However, due to several factors, including the lack of trained healthcare professionals, pervasive underdiagnosis, and stigma, only 0.2% will be able to receive the necessary treatment (Sayers 2001). Hence, key current challenges include the improvement of the efficacy of the diagnosis of psychological disturbances and overcoming known limitations of current clinical scales (Bagby et al., 2004, Zimmerman et al., 2004, Beth & Allerup 1981, Maier, Philipp, & Gerken 1985) together with accurately capturing symptoms and patient specific concerns (Demyttenaere et al., 2020). To this end, we propose that an optimal evaluation strategy may comprise explicit, observer-rated and self-reported evaluation tools combined with implicit physiological and behavioral monitoring using biometric sensing, such as the proposed affective rating methods and associated tools (Reinertsen & Clifford 2018).”

We thank the Editor and the Reviewers again for the care they have taken in processing this manuscript. We hope that you will find that the reworked version of our manuscript complies with the concerns raised in the referee reports. Thank you for considering our work.

Kind regards,

Héctor López Carral and co-authors

References:

Kurdi, Benedek, Shayn Lozano, and Mahzarin R. Banaji. "Introducing the open affective standardized image set (OASIS)." Behavior research methods 49.2 (2017): 457-470.

The Center for Systems Science and Engineering, Johns Hopkins. Coronavirus COVID-19 Global Cases; 2020.

Holmes, E. A., O’Connor, R. C., Perry, V. H., Tracey, I., Wessely, S., Arseneault, L., Bullmore, E. (2020). Multidisciplinary research priorities for the COVID-19 pandemic: a call for action for mental health science. The Lancet Psychiatry. Elsevier Ltd.

Rajkumar, R. P. (2020). COVID-19 and mental health: A review of the existing literature. Asian Journal of Psychiatry, 52.

Torales, J., O’Higgins, M., Castaldelli-Maia, J. M., & Ventriglio, A. (2020). The outbreak of COVID-19 coronavirus and its impact on global mental health. International Journal of Social Psychiatry, 66(4), 317–320.

Hamilton, Max. "The Hamilton Depression Scale—accelerator or break on antidepressant drug discovery." Psychiatry 23 (1960): 56-62.

Williams JB: Standardizing the Hamilton Depression Rating Scale: past, present, and future. Eur Arch Psychiatry Clin Neurosci 2001; 251(suppl 2): II6–II12

Bech P. Fifty years with the Hamilton scales for anxiety and depression. A tribute to Max Hamilton. Psychother Psychosom. 2009; 78(4):202–11

Bagby, R. M., Ryder, A. G., Schuller, D. R., & Marshall, M. B. (2004). The Hamilton Depression Rating Scale: has the gold standard become a lead weight? American Journal of Psychiatry, 161(12), 2163-2177.

Gibbons, R. D., Clark, D. C., & Kupfer, D. J. (1993). Exactly what does the Hamilton depression rating scale measure? Journal of psychiatric research, 27(3), 259-273.

Stefanis, C. N., & Stefanis, N. C. (2002). Diagnosis of depressive disorders: A review. Depressive disorders, 1-87.

Gullion, C. M., & Rush, A. J. (1998). Toward a generalizable model of symptoms in major depressive disorder. Biological psychiatry, 44(10), 959-972.Gullion, C. M., & Rush, A. J. (1998). Toward a generalizable model of symptoms in major depressive disorder. Biological psychiatry, 44(10), 959-972

Shapiro, M. B. (1951). An experimental approach to diagnostic psychological testing. Journal of mental science, 97(409), 748-764.Shapiro, M. B. (1951). An experimental approach to diagnostic psychological testing. Journal of mental science, 97(409), 748-764.

Fava, G. A., & Belaise, C. (2005). A discussion on the role of clinimetrics and the misleading effects of psychometric theory. Journal of clinical epidemiology, 58(8), 753-756.

Zimmerman M, Posternak MA, Chelminski I. Is it time to replace the Hamilton Depression Rating Scale as the primary outcome measure in treatment studies of depression? J Clin Psychopharmacol. 2005 Apr;25(2):105–10.

Beth. P., Allerup, N., Rcisby, N. & Gram, L. F. (1984). Assessment of symptom change from improvement curves on the Hamilton depression scale in trials with antidepressants.

Maier. W., Philipp. M., & Gcrken, A. (1985). Dimensions of the Hamilton Depression Scale.

Fried, E. I., & Nesse, R. M. (2015). Depression sum-scores don’t add up: why analyzing specific depression symptoms is essential. BMC medicine, 13(1), 1-11.

Beck A, Steer RA, Garbin MG. (1988) Psychometric properties of the Beck Depression Inventory: 25 years of evaluation. Clin Psychol Rev; 8:77–100.

Santor DA, Gregus M, Welch A. (2009) Eight decades of measurement in depression. Measurement. 4:135–55.

Kessler, R. C., Berglund, P., Demler, O., Jin, R., Koretz, D., Merikangas, K. R., & Wang, P. S. (2003). The epidemiology of major depressive disorder: results from the National Comorbidity Survey Replication (NCS-R). Jama, 289(23), 3095-3105.

Sayers, J. (2001). “The world health report 2001 - Mental health: new understanding, new hope.” Bulletin of the World Health Organization 79.11, p. 1085.

Demyttenaere, K., Kiekens, G., Bruffaerts, R., Mortier, P., Gorwood, P., Martin, L., & Di Giannantonio, M. Outcome in depression (II): beyond the Hamilton Depression Rating Scale. CNS spectrums, 1-22.

Vigo, D., Thornicroft, G., and Atun, R. (2016). “Estimating the true global burden of mental illness”. The Lancet Psychiatry 3.2, pp. 171–178.

Reinertsen, E., & Clifford, G. D. (2018). A review of physiological and behavioural monitoring with digital sensors for neuropsychiatric illnesses. Physiological measurement, 39(5), 05TR01.

Whelan, R. (2008). Effective analysis of reaction time data. The Psychological Record, 58(3), 475-482.

Taylor, D. A., Altman, I., Wheeler, L., & Kushner, E. N. (1969). Personality factors related to response to social isolation and confinement. Journal of consulting and clinical psychology, 33(4), 411.

Kong, X., Wei, D., Li, W., Cun, L., Xue, S., Zhang, Q., & Qiu, J. (2015). Neuroticism and extraversion mediate the association between loneliness and the dorsolateral prefrontal cortex. Experimental brain research, 233(1), 157-164.

Zelenski, J. M., Sobocko, K., & Whelan, D. C. (2014). Introversion, solitude, and subjective well-being. The handbook of solitude: Psychological perspectives on social isolation, social withdrawal, and being alone, 184-201.

Center for Disease Control, Rothstein, M. A., Alcalde, M. G., Elster, N. R., Majumder, M. A., Palmer, L. I., & Hoffman, R. E. (2003). Quarantine and isolation: Lessons learned from SARS. University of Louisville School of Medicine, Institute for Bioethics, Health Policy and Law.

Nobles, J., Martin, F., Dawson, S., Moran, P., & Savovic, J. (2020). The potential impact of COVID-19 on mental health outcomes and the implications for service solutions.

Lipton, Z. C., Elkan, C., & Naryanaswamy, B. (2014). Optimal thresholding of classifiers to maximize F1 measure. In Lecture Notes in Computer Science (including subseries Lecture Notes in Artificial Intelligence and Lecture Notes in Bioinformatics) (Vol. 8725 LNAI, pp. 225–239). Springer Verlag.

Miles, S. H. (2015). Kaci Hickox: public health and the politics of fear. The American Journal of Bioethics, 15(4), 17-19.

Brooks, S. K., Webster, R. K., Smith, L. E., Woodland, L., Wessely, S., Greenberg, N., & Rubin, G. J. (2020). The psychological impact of quarantine and how to reduce it: rapid review of the evidence. The Lancet.

Hossain, M. M., Sultana, A., & Purohit, N. (2020). Mental health outcomes of quarantine and isolation for infection prevention: A systematic umbrella review of the global evidence. Available at SSRN 3561265.

Barbisch, D., Koenig, K. L., & Shih, F. Y. (2015). Is there a case for quarantine? Perspectives from SARS to Ebola. Disaster medicine and public health preparedness, 9(5), 547-553.

Rubin, G. J., & Wessely, S. (2020). The psychological effects of quarantining a city. Bmj, 368.

Alberto Conti, A. (2020). Historical and methodological highlights of quarantine measures: From ancient plague epidemics to current coronavirus disease (covid-19) pandemic. Acta Biomedica. Mattioli 1885.

Shah, J., Shah, J., & Shah, J. (2020). Quarantine, isolation and lockdown: in context of COVID-19. Journal of Patan Academy of Health Sciences, 7(1), 48-57.

Braun, H. I., Jackson, D. N., & Wiley, D. E. (Eds.). (2001). The role of constructs in psychological and educational measurement. Routledge.

Paulhus, D. L. (2002). Socially desirable responding: The evolution of a construct. The role of constructs in psychological and educational measurement, 49459.

Paulhus, D. L. (2017). Socially desirable responding on self-reports. Encyclopedia of personality and individual differences, 1-5.

---

## [Editor Report · Decision Letter 1]

31 Jul 2020

Subjective ratings of emotive stimuli predict the impact of the COVID-19 quarantine on affective states

PONE-D-20-15454R1

Dear Dr. Carral,

We’re pleased to inform you that your manuscript has been judged scientifically suitable for publication and will be formally accepted for publication once it meets all outstanding technical requirements.

Kind regards,

Stephan Doering, M.D.

Academic Editor

PLOS ONE

---

## [Editor Report · Acceptance letter]

6 Aug 2020

PONE-D-20-15454R1 

Subjective ratings of emotive stimuli predict the impact of the COVID-19 quarantine on affective states 

Dear Dr. López-Carral:

I'm pleased to inform you that your manuscript has been deemed suitable for publication in PLOS ONE. Congratulations! Your manuscript is now with our production department. 

Kind regards, 

on behalf of

Professor Stephan Doering 

Academic Editor

PLOS ONE